# Influence of Weathering on the Degradation of Cellulose Acetate Microplastics Obtained from Used Cigarette Butts

**DOI:** 10.3390/polym15122751

**Published:** 2023-06-20

**Authors:** Branka Mušič, Anita Jemec Kokalj, Andrijana Sever Škapin

**Affiliations:** 1Department of Materials, Slovenian National Building and Civil Engineering Institute, Dimičeva ulica 12, SI-1000 Ljubljana, Slovenia; andrijana.skapin@zag.si; 2Department of Biology, Biotechnical Faculty, University of Ljubljana, Večna Pot 111, SI-1000 Ljubljana, Slovenia; anita.jemec@bf.uni-lj.si; 3Faculty of Polymer Technology—FTPO, Ozare 19, SI-2380 Slovenj Gradec, Slovenia

**Keywords:** cellulose acetate, cigarette filters, microplastics, artificial weathering, polymer degradation

## Abstract

Cellulose acetate is used in many applications, including for cigarette filters. Unfortunately, unlike cellulose, its (bio)degradability is under question, yet it often ends up uncontrolled in the natural environment. The main purpose of this study is to compare the effects of weathering on two types of cigarette filter (classic filters and newer filters that have more recently arrived on the market) following their use and disposal in nature. Microplastics were prepared from polymer parts of used (classic and heated tobacco products—HTP) cigarettes and artificially aged. TG/DTA, FTIR, and SEM analyses were performed both before and after the aging process. Newer tobacco products contain an additional film made of a poly(lactic acid) polymer which, like cellulose acetate, burdens the environment and poses a risk to the ecosystem. Numerous studies have been conducted on the disposal and recycling of cigarette butts and cigarette butt extracts, revealing alarming data that have also influenced the decisions of the EU, who addressed the disposal of tobacco products in the EU Directive (EU) 2019/904. Despite this, there is still no systematic analysis in the literature evaluating the impact of weathering (i.e., accelerated aging) on the degradation of cellulose acetate in classic cigarettes compared with that in newer tobacco products that have recently appeared on the market. This is of particular interest given that the latter have been promoted as being healthier and environmentally friendly. The results show that in cellulose acetate cigarette filters the particle size decreased after accelerated aging. Also, the thermal analysis revealed differences in the behavior of the aged samples, while the FTIR spectra showed no shifts in the position of the peaks. Organic substances break down under UV light, which can be seen by measuring the color change. The PLA film was found to be more stable than cellulose acetate under the influence of UV light.

## 1. Introduction

Microplastics, i.e., plastic particles smaller than 1 mm, have become a global problem in both aquatic and terrestrial habitats. Cigarette filters are also a source of microplastics that is usually insufficiently considered, despite the enormous amount of this type of waste. Roughly 4.5 trillion cigarette filters pollute our oceans, rivers, city sidewalks, parks, soil, and beaches every year [1].

Tobacco products already affect the environment during the growth and cultivation of tobacco, and during the production, distribution, use, and disposal of the finished products, thereby poisoning the aquatic and terrestrial environment [2,3]. Many authors have conducted scientific research monitoring environmental ecotoxicity, tracking the release of nanoparticles from cigarette filters, the amount of carbon released, and the presence or release of certain heavy metals, such as As, Pb, Cd, Cu, Ni, Sr, Ti, Cr, Co, Al, Ba, Mn, Zn, and Fe [4,5,6,7]. In addition to significant amounts of nicotine and heavy metals, used cigarette filters contain other compounds such as hydrogen cyanide, ammonia, acetaldehyde, formaldehyde, benzene, phenol, and pyridine, amongst others [7]. An additional problem lies in the fact that cigarettes contain polymer parts—namely the cigarette butts, made of plasticized cellulose acetate fibers—which, in the long term, degrade into microplastics. The uncontrolled disposal of these plastics into the environment is therefore a particular problem. In the natural environment, these plastics are exposed to various environmental conditions, such as rain, the abrasion of solid particles, salt, temperature variation, and most importantly, UV radiation, which significantly accelerates the degradation of the polymer into microplastics. Some research has, however, already been carried out in this field, investigating the influence of various weather/natural conditions on the deterioration of cigarette filters [8,9,10,11]. The results showed that the rate of release of toxic chemicals increases along with increasing temperatures, both in the environmental chambers and when exposed to outdoor conditions on a roof. Certain chemicals are also excreted more quickly at higher humidity, while water-soluble chemicals migrate more quickly into the environment via water [12]. An investigation into the effect of temperature and UV radiation at different wavelengths on the degradation of cellulose acetate fibers revealed that the greatest changes occurred in cellulose acetate fibers irradiated with UV light at wavelengths greater than 235.3 nm. During the photodegradation of cellulose acetate fibers at room temperature, gaseous products, mainly CO, CO_2_, and CH_4_, were formed, and the proportion of acetic acid and total mass both decreased. Slightly worse mechanical properties were also observed in the irradiated cellulose acetate fibers, such as reduced tensile strength and relative elongation, due to the formation of free radicals causing the splitting of polymer chains and side groups [13]. The degradation of used and unused cellulose and cellulose acetate cigarette filters has also been studied in various different environments, such as soil (on the surface of the earth in nature) or in a composting container [14], and in various aquatic environments, including freshwater, seawater, and artificial seawater [15]. The physical and chemical properties and degradation of unused cigarette filters have also been investigated and compared with artificially smoked cigarettes and filters obtained from a natural environment [16]. The importance of this area has been presented in a comprehensive annual report by the World Health Organization (WHO) [17], as well as in the EU Directive (EU) 2019/904, which outline the need to reduce the environmental impact of certain plastics, including cellulose acetate (CA), from tobacco products [18]. Due to the huge amount of waste cigarette filters, researchers are searching for possibilities to recycle CA filters. The limiting step in their decomposition is the hydrolysis of the CA polymer into cellulose and acetic acid, which is extremely slow under environmental conditions and poses a significant risk to the environment [19]. It is not possible to make CA cigarette filters biodegradable due to the high degree of acetate substitution; this makes the functionalized/modified cellulose inaccessible to microorganisms, thus preventing biodegradability in nature [20].

Even in new cigarette products, plasticized cellulose acetate fibers remain the main component of the filter, but it is necessary to pay particular attention to the newer tobacco products that, in addition to CA, contain polymer films made of poly(lactic acid) polymer (PLA) and thus place an additional burden on the environment. PLA and CA can be biodegraded through chemical hydrolysis and UV radiation, which is a long-term process under environmental conditions. The slow decomposition of cigarette filters in the environment is also associated with low nutrient content, especially with respect to nitrogen, which limits the activity of microbes or microorganisms [21].

In the present work, we studied the effect of UV light from a Xenon arc lamp (which produces the most realistic reproduction of full-spectrum sunlight, including ultraviolet, visible light, and infrared radiation [22]) on classic used cigarette butts and compared it to the impact on used cigarette butts from newer heated tobacco products (HTP) containing both CA and PLA.

## 2. Materials and Methods

### 2.1. Materials

Used cigarette filters/butts were collected in order to prepare two different types of cellulose acetate micro plastics in powder form, namely: (a) powder made from the used butts of conventional cigarettes and (b) powder from cigarette butts used in heated tobacco devices (HTP). All filters collected were separated from the remaining parts of the cigarettes. In all cases, the wrapping paper was removed from the filter, and care was taken to avoid any contamination with tobacco or ash. The filters of HTP contain an additional biodegradable, pleated poly(lactic acid) (PLA) film which serves as a cooling plug, designed for aerosol cooling. This water-insoluble, corn-based polymeric solid will be inert in the environment, and surface degradation is expected due to exposure to sunlight. The content of PLA in a tobacco product is quite high, as the maximum amount of PLA used in this tobacco product is as much as 30.1%, expressed as a percentage of the weight of the product. These PLA filters were also collected, and formed the third sample investigated (c). The cellulose acetate cigarette filters and PLA polymer films were then ground separately in a ball mill to produce a loose powder for each of the three types of filter.

All used cigarette filters from regular/classic cigarettes collected for the purpose of this study were of the same brand, namely BOSS classic cigarettes (by Tobačna Ljubljana d.o.o.). For used cigarette filters from heat tobacco devices (the abbreviation HTP) or the generally known term IQOS (the abbreviation for “I Quit Ordinary Smoking”, by Philip Morris International, Inc., Stamford, CT, USA), the brand HEETS was collected. In continuation, the CA filter samples were designated classic and IQOS, respectively.

Pure CA from Acros Organics was used as a reference CA for FTIR analysis.

### 2.2. Methods

#### 2.2.1. Preparation of Material for Testing

Vibrating ball mills are well-known tools that have been used for decades to reduce the particle size of materials on both laboratory and pilot scales. The processes in vibrational ball mills are complex and strongly depend on the material being processed, the filling of the drum, the filling ratio between the grinding bodies (balls) and the material being ground, and the properties of the grinding bodies themselves (type, quantity, size). For each system, it is necessary to determine the optimal grinding conditions, the grinding time and frequency, as well as the process itself (1-stage, 2-stages).

A vibrational ball mill MixMill MM20 (Domel, Slovenia) was used for the preparation of all the material in the present investigation. Briefly, we performed a two-stage grinding process. In the first step, 0.8–1 g of cellulose acetate cigarette butts were placed in a stainless steel drum with 3 hardened stainless steel balls of 0.8 mm in diameter. The mechanical milling process was conducted at room temperature, at a frequency of 30 Hz for 1 min. In the second stage, the grinding bodies were replaced with 1 ball, 2.5 mm in diameter, and grinding took place for 3 min at a frequency of 30 Hz.

The PLA film from the IQOS cigarettes was ground in the same mill using a one-stage grinding process. A total of 0.8–1 g of the PLA film was placed into the stainless steel drum with 1 hardened stainless steel ball of 2.5 mm in diameter. The mechanical milling process was conducted at room temperature, at a frequency of 30 Hz for 7 min.

Figure 1 and Figure 2 show the stages through which the CA microplastics and PLA film material were prepared for further testing.

Each Figure shows the process divided into three steps; the first step, showing the cigarette butts following collection, the second step, during which the various parts were separated and the paper wrapping removed, and the third step, where the CA filters were milled.

During use, the PLA film of IQOS cigarettes is thermally damaged, as shown in Figure 3. As the thermally damaged parts of the film are very hard, they could not be ground into microplastics. These parts of the PLA film, which remained in larger pieces after grinding, were excluded from further processing, whilst the remainder of the sample (i.e., the ground PLA film) was sieved through a nominal sieve opening 230 meshes (63 µm).

#### 2.2.2. Bulk Density Measurements

The volumetric bulk (apparent) density of the prepared powders at room temperature (23 ± 2 °C) was determined by a pycnometer, which measures the gravimetric density of a powder using a high-precision analytical balance. The powder was added to a standardized Erichsen Mod 290/IV pycnometer (ERICHSEN GmbH & Co., Hemer, Germany), and then weighed, giving the weight of the powder sample at a constant volume.

#### 2.2.3. Particle Size and Particle Distribution

Particle size and distribution were determined by a Microtrac MRB SYNC Analyzer, using a Microtrac Bluewave unit (ERICHSEN GmbH & Co., Hemer, Germany) for wet-method measurements. The samples were measured in three sequentially performed runs, from which averages were calculated and used for further data evaluation.

#### 2.2.4. Thermal Decomposition

Thermal decomposition of the cigarette filter specimens was determined by thermogravimetric analysis (TG/DTA), using a Netzsch STA 409PC Luxx instrument (Weyhe, Germany). Specimens with a mass of approximately 5 mg were heated in airflow from room temperature to 500 °C at a rate of 10 °C/min.

#### 2.2.5. Fourier-Transform Infrared Spectroscopic Analysis of the Milled Samples

In order to determine differences in the used material before and after artificial weathering for each of the three cellulose acetate filters and the PLA film, attenuated total reflection Fourier-transform infrared spectroscopy (FTIR) was carried out using a FTIR Spectrum Two spectrometer (PerkinElmer, Waltham, MA, USA). The spectra were recorded from 400 cm^−1^ to 4000 cm^−1^, with an average of four scans taken at a resolution of 4 cm^−1^.

#### 2.2.6. Surface Morphology

The surface morphologies and the shape and size of the microparticles were observed using a JSM-IT500LV scanning electron microscope (SEM) (Jeol, Tokyo, Japan) with SEM operational software. It has integrated energy dispersive spectroscopy, a W filament, and fully automatic gun alignment. Measurements were performed in low (101–103 Pa) vacuum mode, at a working distance of 10 mm, and an accelerated voltage of 10 kV, using a secondary electron detector (BED-S). The milled cigarette butt samples were placed on double-sided carbon tape.

#### 2.2.7. Artificial Aging and Color Change Measurements

The microplastics obtained were also aged under UV irradiation, according to standard EN ISO 4892-2:2013 [23]. The accelerated weathering of the filter specimens was conducted in a Q-SUN Xe-3 Xenon arc chamber (Q-Lab, Manchester, UK), which reproduces the damage caused by full-spectrum sunlight (noon summer sunlight for 24 h a day). The powder samples were exposed to the Xenon light for 1000 h.

In order to determine the change in color in the ground cigarette filters that occurred as a result of the influence of UV irradiation, an i1 spectrophotometer (X-rite, Grand Rapids, MI, USA) was used to measure the color coordinates of the ground samples in the CIE *L***a***b** color space [24] before and after light exposure in the Xenon chamber. It is possible to see the total color change and the change in *L**, *a**, and *b** values (*L**: lightness, *a**: red/green value, *b**: blue/yellow value) from these results.

All three samples were tested before and after artificial accelerated aging according to the various testing methods described in Section 2.2.

## 3. Results

### 3.1. Bulk Density Measurements

The volumetric bulk density of the prepared CA microplastics and PLA film are presented in Table 1.

The results were calculated based on the measurement of five parallels. As shown in the table, the apparent densities of CA in conventional (classic) and IQOS cigarettes were different, even though the material was prepared and milled under the same conditions. Milled cellulose acetate filters from IQOS cigarettes have a higher bulk density than those from classic cigarettes, which could potentially be attributed to the difference in the preparation process.

### 3.2. Particle Size Distribution

The particle size distribution graph for the ground CA filter powder from classic cigarettes shows two peaks—one at 47.9 μm and a far smaller peak at 4.5 μm (see the blue curve in Figure 4). In total, 80% of the particles were distributed between 30 μm and 100 μm. The particle size was again measured after 1000 h of exposure in the Xenon chamber, at which point the particles were mainly (80%) distributed between 20 μm and 90 μm (shown by the red curve in Figure 4). It was evident that the CA particle size obtained from classic cigarettes decreased as a result of aging in the Xenon chamber. A decrease in the ground CA filter particle size obtained from IQOS cigarettes was also observed, as before light exposure in the Xenon chamber 80% of the particles were distributed between 20 μm and 100 μm, while following the aging process 80% of the particles were distributed between 16 μm and 100 μm. The results are summarized in Table 2.

The particle size distribution of CA from classic cigarette filters is shown in Figure 4 with q3 percentiles (each representing the magnitude of x under which a certain amount of sample was located) in each case.

The particle size and distribution of CA from the IQOS cigarette filters are shown in Figure 5, with denoting q3 percentiles in each case.

This analysis shows that the mean particle sizes (in the maximum differential distribution curves, Figure 4) change little after 1000 h of artificial aging under the conditions described in ISO 4892-2:2013. But the fact that the particle size in itself changes according to the parameters used for the milling process should also be taken into account.

### 3.3. Thermal Decomposition

The CA fibers in the cigarette butt undergo thermal degradation during the heating process, resulting in the release of various volatile organic compounds (VOCs) and the formation of char residue. The weight loss observed during the analysis is primarily due to the release of these VOCs, in addition to the breakdown of the CA fibers [17].

The CA curves of the classic and IQOS filters (Figure 6 and Figure 7) show mass loss and the presence of a broad endothermic event between 30 °C and 200 °C. It has been shown that this may be due to the evaporation of water and VOC [25], as cigarette filters lose mass due to the release of compounds that are trapped in the filters during cigarette (tobacco) use, already during exposure to the environment [16]. From the thermal decomposition of individual tobacco constituents, the constituents which are thermally decomposing in the various temperature regions from used cigarette butts are discovered: at 100–200 °C there is the evaporation of volatile substances and bound water, and initial decomposition of pectin and sugars, while in the temperature range 200–400 °C the decomposition of sugars occurs (lignins, pectin, hemicellulose) [25].

Unfortunately, pronounced glass transition of CA was not observed in the samples, which appears at around 160–180 °C [26], probably due to the disturbance of the other ingredients mentioned above.

In Figure 6, the decomposition of CA appeared between 350 °C and 400 °C, mainly due to a cellulose degradation process such as depolymerization, dehydration, and the decomposition of glucosyl [27,28,29]. We can see that in the case of unaged samples, this temperature is around 350 °C and in the case of aged samples this temperature shifts to a higher value, around 400 °C. The DSC peak of CA of an unaged sample obtained from classic cigarette filters at 460 °C represents the complete thermal decomposition of CA [27,28,29]. We note that the aging of this sample shifts this peak to higher temperatures, about 490 °C.

Differences in the DTA curves of the CA from classic cigarettes pre- and post-aging show the additional changes in the heat flow of the sample, which correspond to the different course of exothermic processes above 300 °C. These differences are evident from the shape of the curves as well as the area beneath them, both of which show a significantly narrower peak in the unaged sample. In the case of the aged sample, both exothermal peaks shift to higher temperatures.

Similar processes, as detected in the thermogravimetric analysis of CA classic cigarette filters, also take place in CA IQOS cigarette filters (Figure 7). The essential difference that can be observed is that the processes move to slightly higher temperatures. Thus, CA degradation occurs in a slightly narrower band, between 350 °C and 370 °C, the same due to the cellulose degradation process, such as depolymerization, dehydration, and glucosyl degradation [28,29]. Also, here we can see that for CA IQOS unaged samples this temperature is around 350 °C, but for aged samples, this temperature shifts slightly to higher values.

As in the case of classic cigarette filters, the last peak belongs to the complete thermal decomposition of CA [27,28,29]. We notice that for CA IQOS unaged cigarette filters, this thermal decomposition moves to a temperature above 500 °C and it is a little bit higher for aged CA IQOS cigarette filters. Here is a smaller difference between the aged and unaged samples as in the case of CA classic cigarette filters (Figure 6).

The endothermic peak of DTA, which absorbed heat around 300 °C and is near to the melting point of CA [26], was also less pronounced than that of the CA sample from classic filters (Figure 6).

The course and temperature of the decomposition of CA cigarette butts depend not only on the conditions and speed of heating but also on the type of CA used (obtained from classic or IQOS cigarettes) and the presence of other additives included in the manufacturing process [30].

It can be seen from Figure 8 that the PLA film loses more than 99 wt. % between 250 °C and 445 °C both pre- and post-aging.

It has been mentioned in the literature that roughly four different areas of mass loss can be seen in CA samples. In our study, degradation primarily occurred between 250 °C and 430 °C in the aged samples and between 150 °C and 430 °C in the unaged samples. Large losses in these areas occur due to the decomposition of organic volatile substances present in saturated filters (tar and nicotine components) [31].

The melting point of PLA is between 150 °C and 170 °C, as can clearly be seen on the blue curve in Figure 8, which shows the mass loss of the unaged sample in this temperature range. When the mass loss is lower, the behavior of the aged PLA sample is more stable. Furthermore, the DTA peaks in the given range are related to the melting of the PLA samples. The initial decomposition temperature of PLA appeared around 250 °C and the ignition temperature at ca. 388 °C [32].

It should be taken into account that a certain amount of VOC had already evaporated from the aged samples during artificial aging [10,16,25]. Hence, the partial difference between the shapes of the curves of the samples before and after aging (the effect of UV decomposition of organic molecules and the influence of temperature on volatility), and as we have already pointed out, although in both cases CA filters were used (IQOS and classic), it is probably the different production processes [30] of classic and IQOS CA cigarette filters that affect the course and temperature of the decomposition of CA cigarette butts.

### 3.4. Fourier-Transform Infrared Spectroscopical Analysis of Samples Fingerprint

Figure 9 shows the FTIR spectra of pure CA manufactured by Acros Organics and CA derived from cigarette filters from both classic and IQOS cigarettes. FTIR spectroscopy confirmed that both classic and IQOS cigarette filters are made of CA.

As shown in Figure 10, after aging in the Xenon chamber, the FTIR spectra of CA filters from classic cigarettes have absorption bands at the same wavenumbers as filters and PLA films from IQOS cigarettes, although the intensities vary. This indicates that the samples did not change chemically during the defined regime of accelerated UV aging.

The FTIR spectra are given without an ordinate scale (i.e., without absorbance values), as the curves completely overlap. A “stacked line” display is shown to demonstrate that the bands are in the same positions.

### 3.5. Surface Morphology

Despite the fact that both the classic and IQOS filters were subjected to the same grinding process, the filter samples are visually different after grinding. The shape of the particles in each of the prepared samples can be seen from the SEM images (Figure 11 and Figure 12). The CA particles ground from classic cigarette filters have a more homogeneous appearance and an elongated (strip) shape, whereas the CA particles ground from IQOS cigarette filters are irregular in shape; the sample primarily consisted of irregularly-shaped aggregates with the presence of some strip-shaped particles.

Differences in the shape of the particles may indicate the use of a different production process for the preparation of cigarette filters, which can also lead to a different course of disintegration, as is clearly evident from the particle size measurements presented in Section 3.2.

### 3.6. Visual Examination Following Artificial Aging

Visual differences between the samples before and after artificial aging (weathering) are presented in Figure 13. CIELab color change measurements were also made and are presented in Table 3.

Color differences were computed as the relative distance between two reference points (i.e., between two mathematically specified colors) within a defined color space (CIELab). This difference is typically expressed as Δ*E* and is calculated by comparing reference and sample *L***a***b** values before and after exposure. Thus, Δ*E* is the overall color change value in the CIELab system. A measurement of the change in color of the samples resulting from aging in the Xenon chamber was calculated from parallels of five measurements of *L**, *a**, and *b** values.

Artificial aging affects the changes that occur on cigarette butts, which can be seen both visually and from measurements of color changes taken from the CIELab system. From Figure 13 and Table 3, it can be seen that the largest color differences occur between the aged and unaged samples of classic CA filters, then between the aged and unaged IQOS CA filter samples, while the smallest difference is for the sample PLA foil. The effect of UV decomposition of organic molecules and the influence of temperature cause the evaporation of VOCs, which can also be attributed to color changes between unaged and aged samples [33,34].

New tobacco products are appearing on the market, which indicates the need for a more organized collection of cigarette filters and thus the protection of nature, as well as further research into the possibility of recycling. In this article, we found that in addition to the CA of classic cigarette filters, the CA and PLA of the new IQOS cigarette filters appearing on the market are UV-unstable. UV instability can be used to help recycle old and newer products, as some research has already shown that UV radiation can accelerate degradation processes, especially when combined with chemical and biodegradation [35,36].

## 4. Conclusions

Microplastics enter the environment in various ways, mostly due to improper waste management, including through the breakdown of larger pieces of plastic products that are dumped in nature [37]. In the case of cigarette butts, it seems that the main problem lies in the fact that cigarette users are not generally aware that neither cellulose acetate nor polylactic acid polymer filters are biodegradable. The collection, sorting, and recycling of these materials is therefore essential to reduce the impact on the environment. Problems, however, arise when there is no adequate method to collect smaller polymer/plastic products, such as cigarette filters/butts. These still mostly end up in the environment where they disintegrate into smaller particles, i.e., the macro product is broken down into smaller particles, which further disintegrate into microplastics and then even further into nanoplastics.

As can be seen from the results of the present study, which investigated two different types of cigarette butts (from classic and IQOS cigarettes), the particle size of all samples decreased following accelerated aging, i.e., after exposure to light in the Xenon chamber. This is supported by the fact that the highest particle packing density occurred when a certain suitable weight fraction of finer particles was present in the mixture, which was due to the effect of artificial aging on the samples exposed to the UV light chamber [38].

No shifts in the position of the peaks were shown by the FTIR spectrums, while thermal analysis revealed only slight differences in the behavior of the aged samples. Considering the relatively short duration of the accelerated aging process compared to the length of time the discarded cigarette butts were exposed to the natural environment, we can conclude that even these minor changes indicate a trend, and that longer exposure times would lead to larger changes in the samples. In addition to the microplastic particles and their mechanical impact, a further problem with used cigarette butts thrown into nature is that substances from tobacco are retained in filters and subsequently leached from the used cigarette butts [39,40,41]. Natural influences, such as UV light and rain, also affect the changes that occur in discarded cigarette butts. Organic substances break down under UV (solar) light [42,43], which is visually apparent following aging in the UV chamber as well as evident from the color change measurements taken from the CIELab system. In order to mitigate the impact of tobacco products on users/people, cigarette manufacturers have added new plastic filters (a PLA film), which, unfortunately, put an additional burden on the environment. Cigarette filters made from PLA are not naturally biodegradable, as specific industrial composting conditions are required for their decomposition [44].

It was observed that the PLA film was more stable than CA cigarette filters under the influence of UV light, with its bulk density changing only slightly, meaning that it will remain in the environment for a longer time before decomposing.

Finally, it should be emphasized that the research presented only covered research on PLA film and CA cigarette filters of classic and novel (IQOS) cigarettes, but we should not forget that other cigarette parts also burden the environment, including tobacco, processing aids (antioxidants, defoamers, finishing oils, and preservatives), additives (pigment, fillers, copolymers), adhesives, the paper that is wrapped around and attached to the filter, and other chemicals used and created in the production of cigarette products.

## Figures and Tables

**Figure 1 polymers-15-02751-f001:**
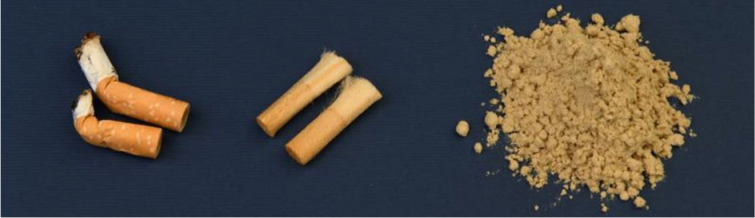
Preparation of the cellulose acetate (CA) sample from classic cigarette butts: (**left**) classic cigarette butts (CA), (**center**) CA without the wrapping paper, and (**right**) milled CA from classic cigarette butts, which served as powder sample (a) in this paper.

**Figure 2 polymers-15-02751-f002:**
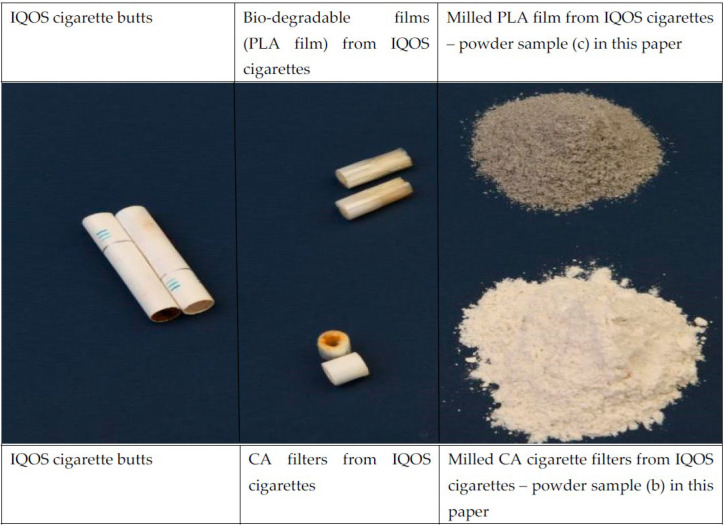
Preparation of cellulose acetate (CA) and PLA film samples from IQOS cigarette butts.

**Figure 3 polymers-15-02751-f003:**
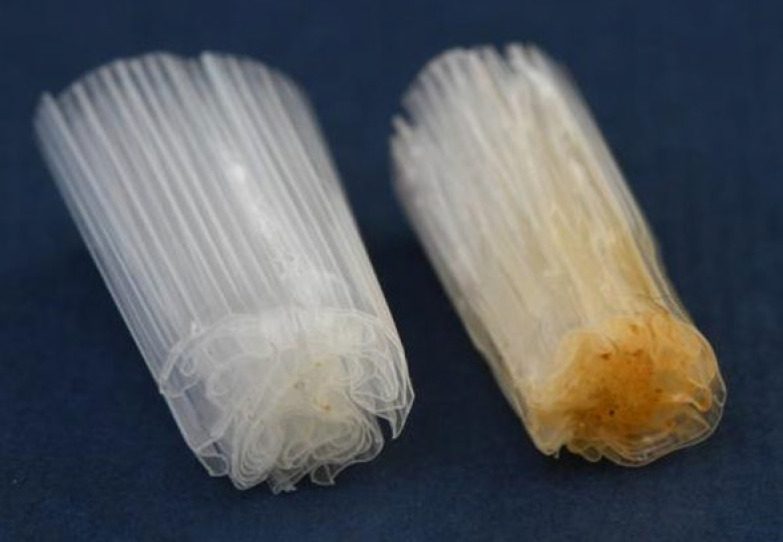
Visual difference between the PLA film of an unused (**left**) and used (**right**) cigarette filter.

**Figure 4 polymers-15-02751-f004:**
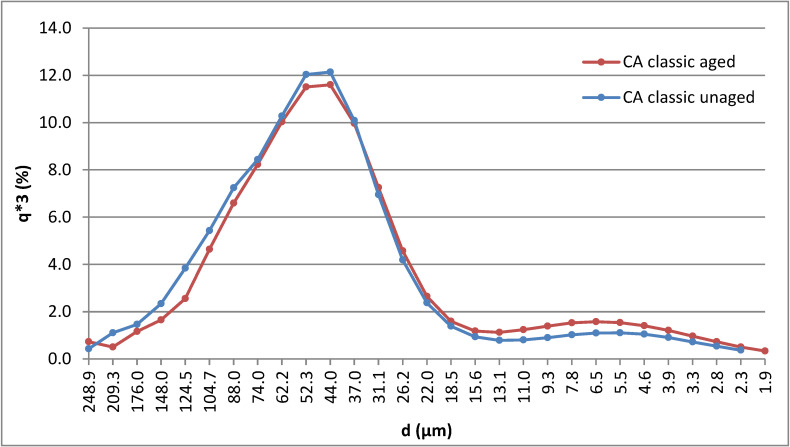
Graphical representation of the particle size distribution of milled CA obtained from classic cigarette filters before (blue curve) and after (red curve) artificial aging.

**Figure 5 polymers-15-02751-f005:**
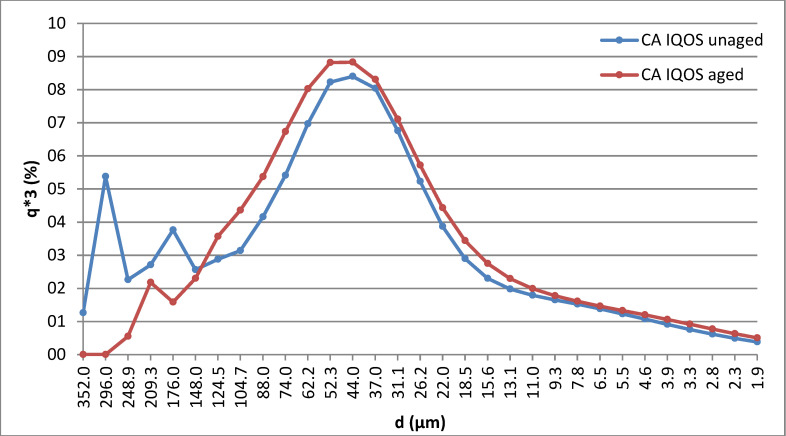
Graphical representation of the particle size distributions of milled CA obtained from IQOS cigarette filters before (blue curve) and after (red curve) artificial aging.

**Figure 6 polymers-15-02751-f006:**
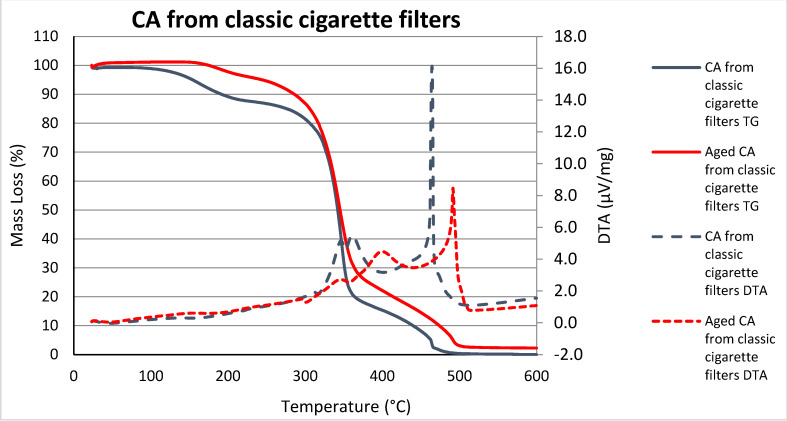
TG/DTA for milled CA filters of classic cigarettes pre- and post-aging.

**Figure 7 polymers-15-02751-f007:**
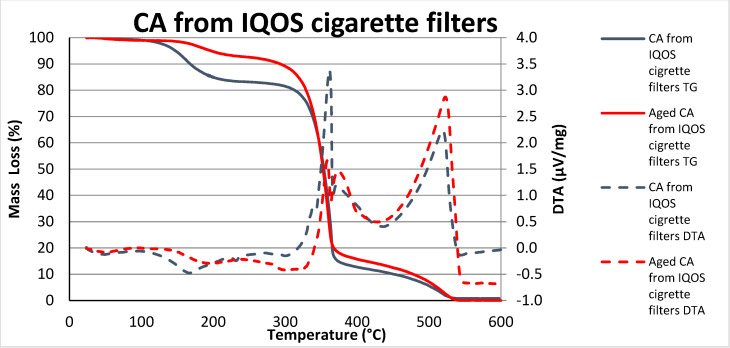
TG/DTA of the milled CA filters from IQOS cigarettes pre- and post-aging.

**Figure 8 polymers-15-02751-f008:**
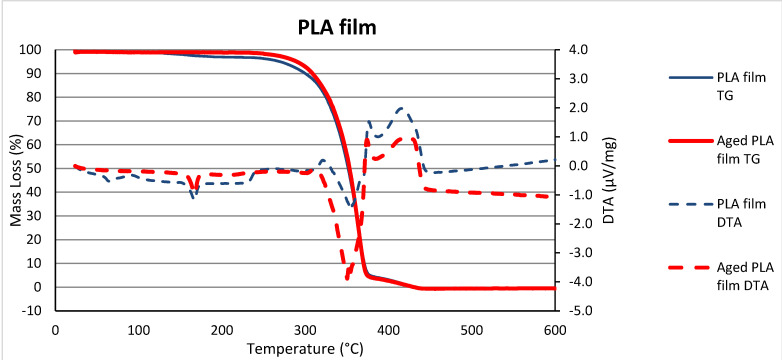
TG/DTA of the milled PLA film from IQOS cigarettes pre- and post-aging.

**Figure 9 polymers-15-02751-f009:**
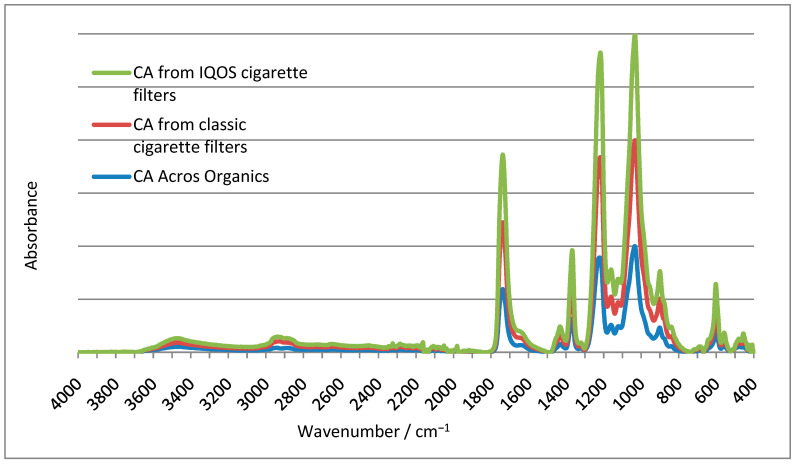
A comparison of the CA obtained from classic and IQOS cigarettes with pure CA (Acros Organics).

**Figure 10 polymers-15-02751-f010:**
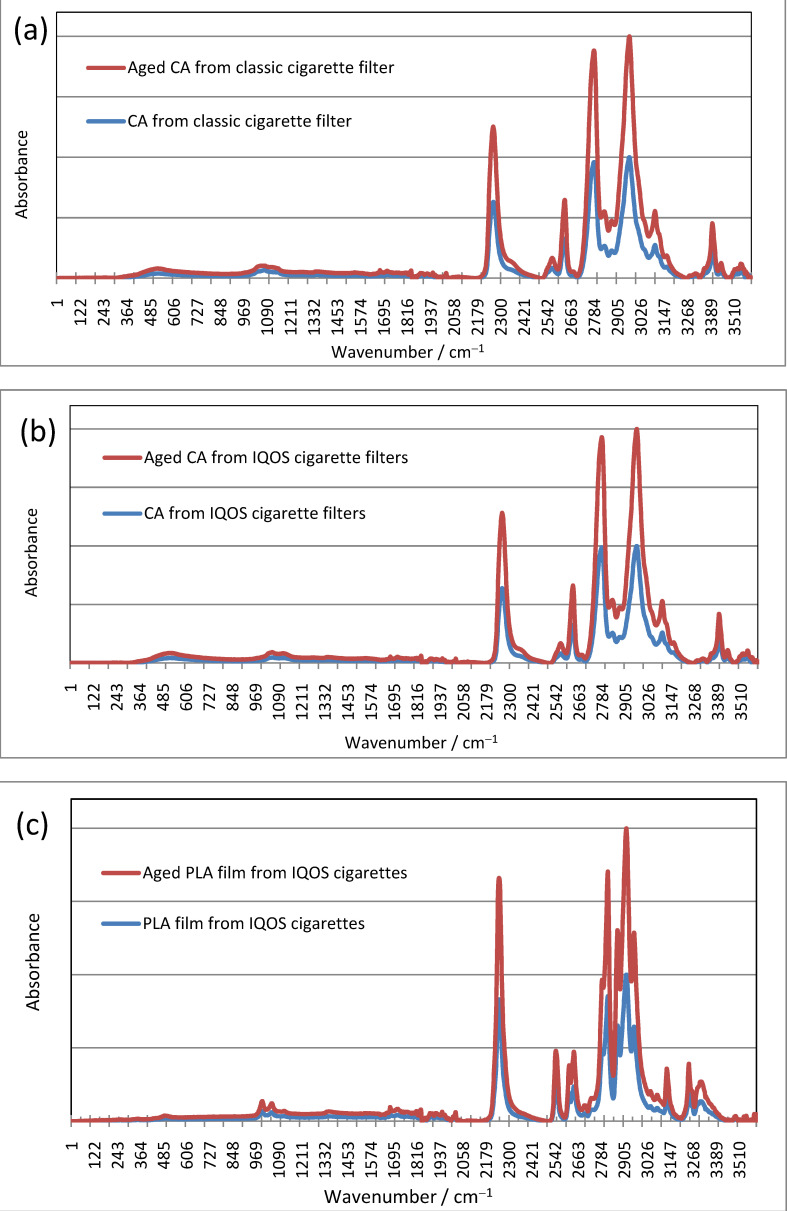
FTIR spectra of (**a**) CA filters obtained from classic cigarette butts, (**b**) CA filters from IQOS cigarette butts, and (**c**) PLA obtained from IQOS PLA film, before and after aging.

**Figure 11 polymers-15-02751-f011:**
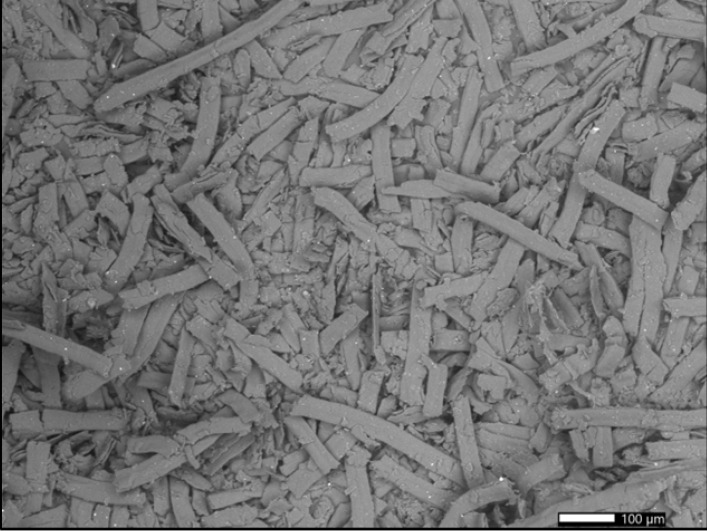
Scanning electron microscope (SEM) micrograph of the surface of milled cellulose acetate (CA) cigarette butts obtained from traditional cigarettes.

**Figure 12 polymers-15-02751-f012:**
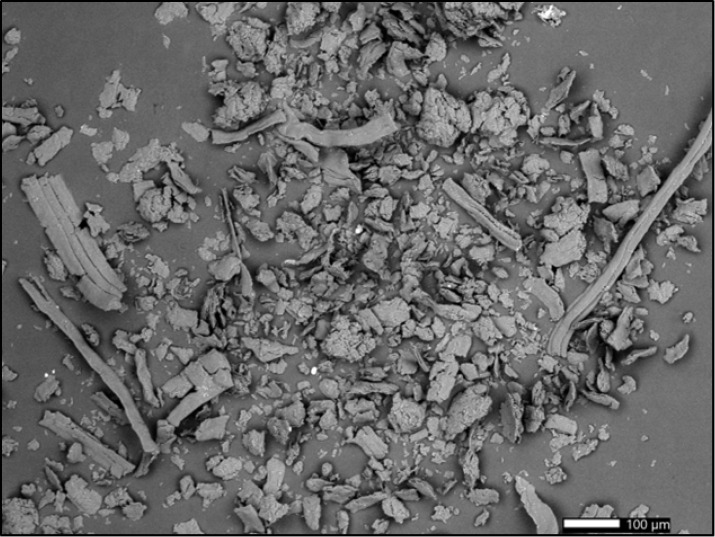
Scanning electron microscope (SEM) micrograph of the surface of milled cellulose acetate (CA) cigarette butts obtained from IQOS cigarettes.

**Figure 13 polymers-15-02751-f013:**
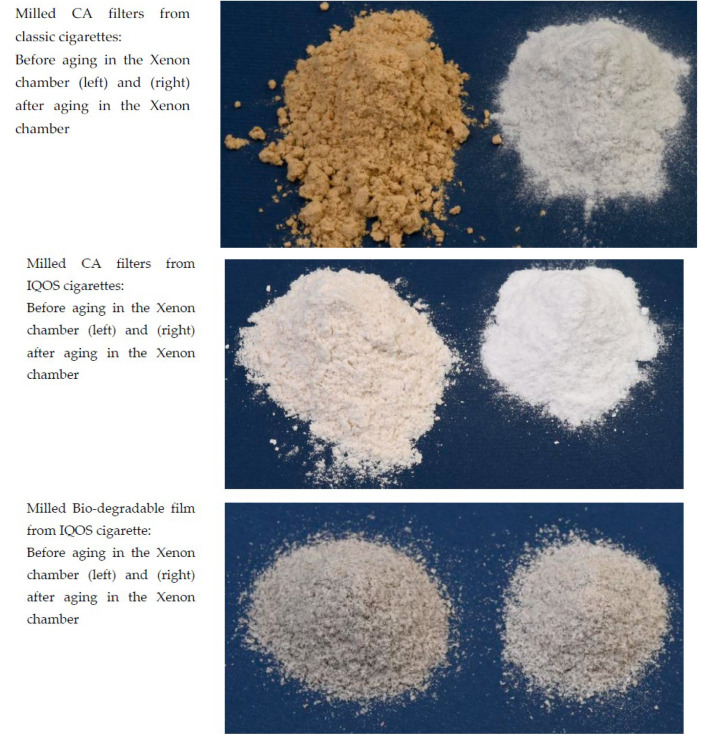
Images to visually represent the samples—the change in color following aging is clearly visible.

**Table 1 polymers-15-02751-t001:** Apparent density of the milled cellulose acetate (CA) microplastics and PLA film. Mean values ± standard deviations are shown.

Milled Filters—Powder Samples	Bulk Density before Weathering (g/100 mL)	Bulk Density after Weathering (g/100 mL)	% Change
(a) CA from classic cigarette	30.1 ± 1	39.0 ± 1	29.6
(b) CA from IQOS cigarette	41.4 ± 1	48.3 ± 1	16.7
(c) PLA film from IQOS cigarette	33.3 ± 1	33.4 ± 1	0.3

**Table 2 polymers-15-02751-t002:** The values of particle size of milled CA obtained from classic and IQOS cigarette filters before and after artificial aging.

Milled CA from Classic Cigarette Filters		Milled CA from IQOS Cigarette Filters	
Before aging	After aging	Before aging	After aging
MV * (µm) = 54.12	MV (µm) = 49.25	MV (µm) = 71.62	MV (µm) = 49.75
MA * (µm) = 25.14	MA (µm) = 20.31	MA (µm) = 21.42	MA (µm) = 18.12
di (%)/d (µm)	di (%)/d (µm)	di (%)/d (µm)	di (%)/d (µm)
10–14.91	10–7.95	10–9.24	10–7.90
20–27.56	20–23.03	20–19.47	20–16.72
30–34.01	30–30.58	30–27.59	30–24.55
40–39.67	40–36.53	40–34.84	40–31.55
50–45.64	50–42.49	50–42.87	50–38.75
60–52.81	60–49.27	60–52.93	60–47.09
70–/	70–/	70–69.32	70–57.72
80–77.13	80–70.46	80–109.7	80–73.82
90–101.6	90–91.57	90–193.4	90–105.5
95–127.2	95–115.3	95–262.7	95–139.3

* MV is the mean diameter, in micrometers, of the “volumetric distribution” and MA is the mean diameter, in micrometers, of the “area distribution”.

**Table 3 polymers-15-02751-t003:** Total color change in the samples following 1000 h in the Xenon chamber.

Milled Filters	CA from Classic Cigarette	CA from IQOS Cigarette	Bio-Degradable Film from IQOS Cigarette
Δ*E*	22.0	16.7	5.2

## Data Availability

Not applicable.

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
