# Peer review of "Influence of Weathering on the Degradation of Cellulose Acetate Microplastics Obtained from Used Cigarette Butts"

_polymers, 2023, doi:10.3390/polym15122751_

Round 1

Reviewer 1 Report

This manuscript is devoted to the study of an important problem associated with the formation of microplastics during the decomposition of cellulose acetate and bioplastic (PCL) from cigarette butts and filters. However, while reading this work, some shortcomings were found.

3.1. Bulk density measurements. Lines 210-211. ...the apparent densities of CA differ from that of conventional and IQOS cigarettes... Remark: This phrase is unclear and should be corrected, e.g., as follows, “the apparent densities of CA in conventional (classic) and IQOS cigarettes were different”

3.2. Particle size distribution.  Remark: Since Figures 5 and 7 repeat data from Table 2, these figures should be deleted as excessive.

Lines 252-255. Discussion on particle size change. Remark: I recommend adding to this discussion that average particle sizes (in a maximum of differential distribution curves, Fig. 4 & 5) do not change (or change very small) after artificial aging.

In addition, the typos and English grammar should be corrected.

Line 85. Remark: Two repetitions “due to”. Remove one.

Line 93. ...a low nutrient... Remark: Remove the article “a” before “low”.

Line 163. ...a high precision... ... Remark: Hyphen is required, ... a high-precision...

Line 180. ....diamond crystal. Remark: Seems that “diamond crystal” is a misprint that can be removed.

Line 191. Accelerated... ... Remark: Article “The” is required before “accelerated”, “The accelerated”...

Line 265. ...post aging... Remark: Hyphen is required, ...”post-aging”...

Line 266....depolymerisation... Remark: ...”depolymerization”... should be written.

Author Response

Dear Reviewer.

Thank you.

Best regards, Branka Mušič

Reviewer 2 Report

This paper studied the effects of weathering (UV exposure) on ground cigarette filter materials.  

1.       Line 13: “it is not (bio)degradable”. Is this statement general enough or only suitable for the cigarette filter type CA?  I understand the definition of bio-degradability is still under debate, but it might be better to be more cautious about this kind of statement or even provide more background. Moreover, it seems that the biodegradability of CA is highly dependent on the degree of acetylation and other factors.

2.       Line 29: what is meant by “old samples”, do you mean samples after aging?

3.       This paper studied a new type of cigarette butts with PLA film. I am quite curious, other readers might do the same, about the functions of this PLA film. Can the authors provide more background regarding this?

4.       Section 3.3 thermal decomposition: it seems that all the aged samples show better thermal resistance according to TGA. Can the authors elaborate on this? Is it because during the aging a lot of small organic molecules have already volatilized? Moreover, can the author provide more exhalations regarding the evident differences between pre- and post-aging samples regarding the shapes of the curves?

5.       It seems the authors only provided explanations regarding the color changes in the conclusion section, wouldn’t it be better to discuss this earlier?

Author Response

(The authors gave the same response as above.)

Reviewer 3 Report

This is only a topical study about the weathering of cellulose acetate and biofilms used in classic and IQOS cigarettes. Few major changes are suggested as follows:

1) Introduction provides unnecessary details about cigarettes and how they pollute the environment. Some of these details could be removed. Please focus only on the pollutant investigated in this study, i.e., cellulose acetate filters.

1.1) On the other hand, previous reports about the recovery, recycling, or biodegradation of cigarette filters are not cited. A more thorough review of previous efforts on recycling or biodegradation of cigarette filters must be provided in order to highlight the novelty of this manuscript.

1.2) Articles reporting about the weathering of any kind of cellulose acetate polymers could be used for comparison (not only about cigarette filters, but also photography films, RO membrane, etc.).

2) The abbreviation for IQOS (I Quit Ordinary Smoking) should be provided the first time it is mentioned, even if it sounds unscientific.

2.1) Also, IQOS is a brand name developed by Philip Morris International, Inc., however these branded cigarettes were not studied. So, why use an abbreviation developed by some multinational corporation to represent a whole class of heated tobacco products?

3) The corn-based cigarette filters are described as PLA in the introduction and then vaguely as biofilms. Could you please use one consistent terminology for the biofilm through the manuscript?

3.1) Additional details, such as chemical name/ composition, has to be provided for the biofilm filters.

4) Is table 2 necessary, since so many charts about the particle size changes are also provided?

5) Figures 4 and 5 could be combined. Similarly, Figures 6 and 7 could be combined.

6) Given that there is a bleaching effect during the artificial aging process (Figure 15),  it is surprising that the FTIR spectroscopy (Figure 12) did not detect the corresponding chemical changes.

Author Response

(The authors gave the same response as above.)

Round 2

Reviewer 2 Report

1.       It seems that the TGA results are still not well explained. Can the authors give more details/reasons about the mass loss before 300 Celsius (which is ~20% for original CA)? Can the authors provide more information about the depolymerization?

2.       Line 325 to 326:” It should be taken into account that a certain amount of VOC evaporates from the 325 aged samples already during artificial aging.” Can the authors add evidences or references to support this?

3.       This paper emphasized the importance of recycling CA filters multiple times. Therefore, can the authors elaborate on the correlations between their findings and the recycling of CA filters? What the readers can learn from their findings?

Author Response

Regards, Branka

Reviewer 3 Report

1) Terminologies: Instead of using the colloquial term IQOS, it will be better to use "HTPs", provide the abbreviation in the abstract and in the introduction, and then include a short explanation (e.g., HTPs are tobacco products that require the use of an electronic device to heat a stick or pod of compressed tobacco) in the materials section. 

2) Section 2.2.4: Did the TGA operate at a rate of 10 oC/min or 10 K/min? Please check the temperature unit.

3) There is definitely a change in the polymer chain size of CA used in classic cigarette butts after aging, as see in the DTA chart of Figure 6. However, this observation is glossed over. Specifically, in lines 281-283, please explain the possible depolymerization/dehydration mechanism of CA, happening at each major DTA peak in Figure 6, based on previous research.

4) Even in CA recovered from IQOS/HTPs, there is a change in the ratio of mass loss at 350 oC versus 500 oC between the original and aged samples (Figure 7, DTA chart). What could this indicate? Please hypothesize based on previous reports.

5) Usually, it is sensible to analyze FTIR spectra using PCA (principal component analysis) in order to glean minor differences between sample sets such as those presented here. Could you possibly analyze your sample spectra via PCA before concluding that no chemical changes could be detected?

6) Please provide references for your statement in lines 398-399.

7) The first paragraph of the Conclusion section could be honestly used as the first paragraph of the Introduction; it is more fitting than the current introduction.

8) The newly added conclusion paragraph (lines 446-451) could be removed because it is not relevant to the presented results.

9) Could you please estimate how much cigarette butt filters are being discarded in to environment, globally, per year? I am sure this information could be extrapolated from the cigarette sales volume and the amount of filter per cigarette. This reviewer wants further justification about the impact of this study in the Introduction section. 

Please avoid informal language. E.g. Line 122- "...to produce a fluffy powder". Instead of fluffy, you could use "loose".

Author Response

Regards, Branka

Round 3

Reviewer 3 Report

The authors have adequately responded to this reviewer's suggestions. The manuscript may be accepted after a minor grammar and spell check.

A minor grammar check is required for the newly revised sections.